# The Management of Phaeochromocytomas and Paragangliomas in the Era of Precision Medicine: Where Are We Now? Evidence-Based Systemic Treatment Options and Future Cluster Oriented Perspectives

**DOI:** 10.3390/ph17030354

**Published:** 2024-03-08

**Authors:** Alessandra Bracigliano, Antonella Lucia Marretta, Luigi Pio Guerrera, Roberto Simioli, Ottavia Clemente, Vincenza Granata, Anita Minopoli, Giuseppina Della Vittoria Scarpati, Fernanda Picozzi, Lucia Cannella, Antonio Pizzolorusso, Francesca Di Gennaro, Roberto Tafuto, Maria Rosaria Sarno, Ernesta Cavalcanti, Dario Ribera, Salvatore Tafuto

**Affiliations:** 1Nuclear Medicine Unit, Istituto Nazionale Tumori I.R.C.C.S. Fondazione “G.Pascale”, 80131 Naples, Italy; a.bracigliano@istitutotumori.na.it (A.B.); f.digennaro@istitutotumori.na.it (F.D.G.); 2Medical Oncology Unit, Ospedale Ave Gratia Plena, San Felice A Cancello, 81027 Caserta, Italy; antomarretta@live.com; 3Sarcoma and Rare Tumors Unit, Istituto Nazionale Tumori I.R.C.C.S. Fondazione “G.Pascale”, 80131 Naples, Italy; luigipioguerrera@hotmail.it (L.P.G.); roberto.simioli@istitutotumori.na.it (R.S.); ottavia.clemente@istitutotumori.na.it (O.C.); fernanda.picozzi@istitutotumori.na.it (F.P.); l.cannella@istitutotumori.na.it (L.C.); a.pizzolorusso@istitutotumori.na.it (A.P.); s.tafuto@istitutotumori.na.it (S.T.); 4Radiology Division, Istituto Nazionale Tumori I.R.C.C.S. Fondazione “G.Pascale”, 80131 Naples, Italy; v.granata@istitutotumori.na.it; 5Laboratory Medicine Unit, Istituto Nazionale Tumori-IRCCS “Fondazione G. Pascale”, 80131 Napoli, Italy; a.minopoli@istitutotumori.na.it (A.M.); e.cavalcanti@istitutotumori.na.it (E.C.); 6Division of Neurosurgery, Department of Neurosciences and Reproductive and Odontostomatological Sciences, Università Degli Studi Di Napoli Federico II, 80131 Naples, Italy; rob.tafuto@gmail.com; 7Division of Pharmacy, Istituto Nazionale Tumori I.R.C.C.S. Fondazione “G.Pascale”, 80131 Naples, Italy; mr.sarno@istitutotumori.na.it; 8Medical Oncology Unit, Sessa Aurunca, Ospedale San Rocco, 81037 Caserta, Italy; darioribera@hotmail.it

**Keywords:** phaeochromocytomas, paragangliomas, neuroendocrine tumors, molecular clusters, management, cluster-specific approach

## Abstract

Pheochromocytomas (PCCs) and Paragangliomas (PGLs), commonly known as PPGLs to include both entities, are rare neuroendocrine tumors that may arise in the context of hereditary syndromes or be sporadic. However, even among sporadic PPGLs, identifiable somatic alterations in at least one of the known susceptibility genes can be detected. Therefore, about 3/4 of all PPGL patients can be assigned to one of the three molecular clusters that have been identified in the last years with difference in the underlying pathogenetic mechanisms, biochemical phenotype, metastatic potential, and prognosis. While surgery represents the mainstay of treatment for localized PPGLs, several therapeutic options are available in advanced and/or metastatic setting. However, only few of them hinge upon prospective data and a cluster-oriented approach has not yet been established. In order to render management even more personalized and improve the prognosis of this molecularly complex disease, it is undoubtable that genetic testing for germline mutations as well as genome profiling for somatic mutations, where available, must be improved and become standard practice. This review summarizes the current evidence regarding diagnosis and treatment of PPGLs, supporting the need of a more cluster-specific approach in clinical practice.

## 1. Introduction

Pheochromocytomas (PCCs) and Paragangliomas (PGLs) are rare neuroendocrine tumors (about 0.2–0.8 new cases per 100,000 people each year) originating from the chromaffin cells derived from the neural crest. PCCs and PGLs (commonly known as PPGLs, to include both entities) are usually slow-growing, mostly catecholamine (CMN) producing tumors, with 30–40% being hereditary [1,2,3,4]. 

The distribution by gender does not show significant differences, although a slight predilection for females has been observed. They are most commonly diagnosed within the third and sixth decades of life, even if they can arise over a wide age range [2,3,5]. Since they usually exhibit indolent growth patterns, they can be detected incidentally in imaging techniques [6].

PCCs originate from the cells of the adrenal medulla, a bilateral organ, located on the top of each kidney, which is the greatest sympathetic paraganglion of the human body. On the other hand, PGLs derive from extra-adrenal paraganglia: thus, they may be located anywhere in the body, and they are commonly classified as sympathetic (thorax, abdomen, pelvis) or parasympathetic (predominantly head and neck, but even mediastinum) depending on their origin.

Regardless of their development background, approximately 85% of paragangliomas arise in the abdomen, while 12% of them in the anterior chest area, and finally about 3% in head and neck. The latter ones, commonly denoted as Head-and-Neck (HN) PGLs, derive exclusively from parasympathetic paraganglia: they are generally asymptomatic and less likely to be metastatic than PGLs of other sites [1,2,3,4,5,6]. 

In this review, we summarize the current evidence regarding management of PPGLs, supporting the need of a more cluster-specific approach in clinical practice in the era of precision medicine.

## 2. Genetics and Molecular Classification of PPGLs: How They Affect Phenotype

PPGLs may be “apparently” sporadic (without any known genetic alterations) or arise in the context of hereditary syndromes (30–40%) with germline mutations in one of the currently identified PPGLs susceptibility genes. However, even sporadic PPGLs (almost 40–50% of all PPGLs) may bear identifiable somatic alterations in at least one of them [5,6,7,8,9,10,11].

This means that around 3/4 of PPGLs patients can be assigned to one well defined molecular cluster, further confirming the importance of genetic testing for germline mutations for every patient in PPGLs management and encouraging somatic tumor mutation analysis as well, although this is still the subject of ongoing research [10,11].

However, limitations include the high costs of genetic testing and the long waiting period required for a diagnosis.

Based on different pathogenetic mechanisms, three main molecular clusters have been identified in the last years: the pseudohypoxia-associated cluster 1, the kinase signaling-associated cluster 2 and Wnt signaling-associated cluster-3 [10,11,12,13,14].

Assigning a particular patient to one of these clusters may guide biochemical testing, the choice of specific imaging modalities, may suggest different clinical behavior and long-term prognosis, and eventually appropriate tailored treatments [10,11], thus rendering the management of PPGLs more personalized.

However, a cluster specific-approach for the treatment of inoperable/metastatic disease has not yet fully entered routine clinical practice.

### 2.1. Cluster 1

The pseudohypoxia-associated cluster is characterized by the activation of several pathways mimicking hypoxia signaling and can be divided into two subclusters (1A, 1B) [10,11,12].

Cluster 1A groups tumors with predominantly germline mutations in Krebs-cycle-related genes such as genes encoding succinate dehydrogenase subunits (SDHx, A, B, C, D), succinate dehydrogenase complex assembly factor-2 (SDHAF2), malate dehydrogenase (MDH2) and fumarate hydratase (FH), while cluster 1b include tumors with either germline or somatic mutations in VHL and EPAS genes [10,11,12,13,14,15].

Germline mutations in genes encoding the SDH subunits A, B, C, and D are associated with the most common hereditary syndromes of PPGLs, formerly referred to as familial pheochromocytoma-paraganglioma syndromes (PGL) type 5, type 4 (PGL4), type 3 (PGL3), and type 1 (PGL1) respectively [9,10,11,14], while germline mutations in the VHL gene are related to Von-Hippel Lindau disease. Less frequent familial forms are associated with germline mutations in MDH2, SLC25A11 and FH genes [9,10,11,12,14,15].

Cluster 1 tumors include about 25–35% of all PPGLs and almost 50–60% of all metastatic PPGLs: they are mainly located at extradrenal locations, particularly SDHx-related tumors [11,12,16,17] while EPAS1-, FH- and VHL- related tumors may occur both as PGLs and PCCs [11,12].

Most of the tumors belonging to this cluster have a noradrenergic biochemical phenotype (norepinephrine -NE- and its main metabolite normetanephrine—NMN) but may also produce dopamine (DA) and its main metabolite 3-methoxytyramine (3-MT), with normal or near normal levels of NE/NMN [9,10,11,12,18]. Of note, in some cases, particularly for SDHx-mutated tumors and HNPGLs, no catecholamine production can be detected and these tumors may be non functional [11,17,18].

Cluster 1-PPGLs, especially those harboring mutations in SDHx genes, are often aggressive, multiple, metastatic with a poorer prognosis [9,10,11,12].

### 2.2. Cluster 2

The kinase signaling-associated cluster includes tumors harboring either germline or somatic alterations in genes which modulate the PI3K/AKT/mTORC1 and RAS/RAF/ERK pathways [10,11,12,13,14].

Genes belonging to this cluster include the RET gene and the NF1 gene which are associated with Multiple Endocrine Neoplasia type 2 (MEN2) and with neurofibromatosis type 1 respectively [13,14,19,20]. 

Cluster 2 includes approximately 50–60% of all PPGLs and almost 2–4% of all metastatic PPGLs [10,11,12]. These tumors are predominantly located adrenally (PCCs) with a typical adrenergic biochemical phenotype (epinephrine/metanephrine-E/MN-> both E/MN and NE/NMN) [9,10,11,12,21]. 

The majority of cluster 2—related PPGLs usually have a better prognosis and low metastatic potential.

### 2.3. Cluster 3

Wnt signaling-associated cluster include tumors that harbor exclusively somatic alterations in genes regulating the Wnt and Hedgehog pathways [10,11,12,13,14].

They represent approximately 5–10% of all PPGLs and they are typically associated with higher chromogranin A expression. Their catecholamine phenotype is still undefined.

Similarly to cluster 1 tumors, these neoplasms typically show aggressiveness as well as early metastatic spread [10,11,12].

## 3. Clinical Presentation and Metastatic Potential 

Clinical presentation of PPGLs is extremely variable and may depend on several factors such as tumor location, size, local extension and/or distant involvement, the secretion or not of catecholamines, the malignancy potential and aetiology (hereditary or sporadic nature) [21].

Most of PPGLs, those deriving from sympathetic paraganglia, generally produce an excessive amount of CMNs [11,21]. 

The release of catecholamines in excess can cause typical adrenergic and noradrenergic symptoms: the most frequent include severe headache, sweating, palpitations, tremor, persistent (cluster 1) or episodic hypertension (cluster 2); therefore, untreated PPGLs, can cause severe damage, particularly to the cardiovascular system [2,6,13,22,23,24,25]. Cluster-specific differences have been reported with cluster 1 patients showing lower basic symptoms scores compared to cluster 2 patients who typically present with paroxysmal symptoms. 

Though rare, about 15% of the paragangliomas (those derived from parasympathetic chains of the head and neck region, HNPGLs) have been traditionally classified as “non-functional” or biochemically “silent”/“pseudo-silent” tumors since they usually do not produce CMNs: common adrenergic-noradrenergic symptoms do not appear and therefore they are harder to detect. As a consequence, they are incidentally found on imaging studies done for other purposes or they may be diagnosed due to their compressive effect on the surrounding structures when they reach a large size [10,11,17,21].

### Metastatic Potential 

All PGGLs are malignant and potentially metastatic and there are no clear-cut features that can predict metastatic behavior: this explains why since the 4th edition of the WHO, PPGLs have no longer been classified as benign and malignant [21,26].

Metastatic disease develops in 15–17% of all PPGLs with different metastatic potential in PCCs (10%) and PGLs (35–40%) [5]. Parasympathetic PGLs (HNPGLs) show typically a very low risk of malignant behavior (about 5%) [10,21,27,28]. 

Estimating the metastatic risk of these tumors is not feasible with histology alone and remains an unmet need. Several scoring systems either based solely on morphological parameters (PASS score, applicable only to PCCs) or combining morphological, immunohistochemical and analytical features (GAPP scale and COPPS score for both PCCs and PGLs) have been proposed during the last years to predict the malignant potential and identify more aggressive lesions. However, the WHO does not endorse but also does not discourage their use [21,29]. 

According to 2021 ESMO Guidelines, a “high risk of metastases” can be established when any patient presents with one or more of the following criteria: (A) tumor size ≥ 5 cm; (B) any extra-adrenal PGL; (C) known SDHB germline mutation; or (D) plasma 3MT > 3 fold above the upper limit of normal [21,30].

## 4. Diagnosis and Staging 

Diagnosis of PPGLs is clinical, biochemical, and radiological. 

Guidelines accept word widely both plasma-free and urinary fractionated metanephrines (MNs) for the screening and follow-up of PPGLs, although recent studies demonstrated a higher sensitivity for plasma over urinary tests [11,21].

False positive results may be related to several conditions such as inappropriate preclinical preparation (diet, incorrect blood sampling), the use of medications that interfere with CMN metabolism, physical stress and laboratory errors. In these cases, tests must be repeated under optimized conditions [10,11,21]. False negative results, despite being less frequent, may be related to small and/or asymptomatic tumors and cluster 1 tumors (DA-producing PPGLs, biochemically silent PGLs of the head and neck region or SDHB mutated-neoplasms) [11,21].

The assessment of plasma 3MT may be useful in cluster 1 tumors [10,11], while the measurement of chromogranin A is strongly recommended in biochemically silent PGLs, particularly in those SDHB-mutated [10,21].

In patients with altered screening examinations, radiological tests are particularly important to detect and stage the tumorin order to plan the subsequent therapeutic approach.

Current evidence suggests that CT is generally more useful than MRI for initial localization of PPGLs in biochemically positive patients because it is widely available, cost-effective, and offers better spatial resolution than MRI [21,24,25,27,31]. 

Radiologically, PPGLs are usually solid, hypervascular, and well-circumscribed masses, with sizes varying from 1 to 15 cm. Larger tumors may exhibit central necrosis, while smaller ones typically display a uniform appearance. Moreover, certain PPGLs may simulate adenomas due to macroscopic fat or may exhibit high attenuation due to hemorrhage or calcifications [6,14,21,24,25,27,31] (Figure 1).

For the screening of PCCs, CT scan shows a high sensitivity (almost 100%) but a low specificity (50%) [11]. On CT scan, PCCs are generally more than 10 Hounsfield units (HU) with a marked enhancement that can be heterogenous due to degenerative changes [12,31] (Figure 1). Differential diagnosis with the less common lipid-poor adenomas may be challenging as they also have more than 10 HU and show similar absolute and relative percentages washout values [31].

MRI has similar sensitivity but higher specificity than CT for PCCs and is superior to CT imaging for screening purposes of extra-adrenal PGLs, especially those of the head and neck region [11]. 

Moreover, being free of ionizing radiations, it must be preferred for initial tumour localization in children (10–20% of all PPGLs diagnoses) and pregnant women as well as for life-long follow-up [11,21].

Typically, PPGLs show a high signal intensity on T2-weighted images (the characteristic “light bulb” bright sign), with low signal intensity on T1-weighted images. However, other lesions can also present with a “light bulb” feature including cysts, lipid-poor adenomas and metastatic lesions and the signal intensity on T1 may increase in the presence of fat or haemorrhage [11,21,31,32].

The use of dynamic contrast enhanced MRI (DCE-MRI) in atypical cases of adrenal masses should be taken into account as its findings may be contributory to the results of chemical shift MRI [33,34]. Furthermore, DCE-MRI showed to be particularly useful in the diagnosis of HNPGLs, as their early initial avid enhancement is distinctive from other cervical benign lesions [32,35,36].

In case of atypical pheochromocytomas that undergo cystic degeneration, integrating MRI findings with those of correlative planar/hybrid radionuclide images should be considered as tumour accumulation of MIBG and/or FDG in the residual solid tissue or in the peripheral tumour rim may help to classify properly these lesions and disclose their real nature [37,38].

In all patients with PPGLs, except for those with low risk of metastatic disease, whole-body CT or MRI must be performed before any surgical approach to rule out metastases or multiplicity [10,11].

Functional imaging is recommended in case of inconclusive results on anatomic imaging as well as for staging in patients with suspected metastatic disease as well as multifocal disease [11,30].

The choice of the most sensitive technique for each case relies on the clinical as well as biochemical profiles, and location of the primary tumour, which are also predictors of the underlying genotype [21]. 

PPGLs cells express on their surface different molecular targets which allow images to be obtained. These include several transporters with different capture mechanisms (Norepinephrine, glucose and amino acid transporters for 123/131 I-MIBG, 18F-FDG or 18F-DOPA respectively) or membrane surface receptors (somatostatin receptors, SSTRs, for 68GA-DOTA-SST analogs) [21,31,32,36].

123/131 I-MIBG are both captured by the norepinephrine transporter: 123I MIBG SPECT/CT sensitivity and specificity is extremely high for sporadic PCCs, though it dramatically decreases for PGLs (52–75%), especially for HNPGLs (18–50%) and for SDHx-related tumours [21,31,32,36,39,40] (Figure 2A–C). 

68Ga-DOTA-SSAs are somatostatin analog radiotracers that bind to SSTRs. 68Ga-DOTA-SSA PET/CT is proving increasingly useful for the diagnosis of certain PPGLs (per-lesion sensitivity > 90%): in particular, it showed excellent sensitivity for detecting both primary tumors and metastatic lesions in cluster 1 mutation carriers (overall detection rate of 98.6% in SDHB-mutated PPGLs) and it is considered the functional image of choice for HNPGLs [21,27,31,32,36] (Figure 2D–F).

18F-DOPA is a radiotracer that accumulates specifically in some PPGLs due to increased uptake by amino acid transporters: 18F-DOPA PET/CT represents the functional imaging of choice in cluster 1B and cluster 2 PPGLs, as well as in FH mutated and polycythemia-related PPGLs [11,21,32,36]. In these cases, if not available, I123-MIBG SPECT/CT may be considered [21].

Of note, a recently published prospective, single-institution study comparing 18 F-FDOPA PET/CT with I123-MIBG SPECT/CT demonstrated that 18F-FDOPA has noninferior sensitivity as well as similar specificity to 123I-MIBG SPECT/CT in the diagnosis of PPGLs, but it is more sensitive in the assessment of recurrence and metastases, further supporting its use in high risk PPGL in general and in those harboring the above mentioned mutations [41].

Finally, 18F-FDG PET may be useful in metastatic disease in general and as functional imaging of second choice in patients harbouring SDHx mutations (cluster 1A) and in cluster 1B tumours [21,31,32,39,42].

Besides staging, when facing metastatic disease, functional imaging play an important role in the management of patients for whom radiometabolic treatment is being considered: I123-MIBG SPECT/TC, if feasible, and 68Ga-DOTA-SSA PET/CT are used to assess the affinity of the tumor for the available radiotracers in order to select the most proper radiopharmaceutical for each patient [21,31].

## 5. Therapeutic Options

A multidisciplinary approach by an expert medical team is needed in order to define the most proper, personalized treatment plan based on both patient characteristics (age, health status) and tumour features (location, size, extent, biochemical profile, functional imaging and eventually genetic background), weighing the pros and cons of each option [14,21].

Surgery represents the mainstay of therapy for the majority of localized PPGLs, whenever possible.

In advanced and metastatic disease, treatment options include symptomatic therapy (alpha-blockers, beta-blockers, catecholamine synthesis inhibitors) for functional PPGLs in order to prevent life-threatening events, systemic medical treatments such as chemotherapy, targeted therapies, SS analogs (SSAs), radiometabolic therapy as well as loco-regional procedures (cytoreductive surgery, external beam radiotherapy, arterial embolization, cryotherapy, RF ablation) [10,11,21,25,30].

### 5.1. Surgery

Radical surgical resection is the cornerstone of the treatment for the majority of localized PPGLs as it remains the only potentially curative therapeutic option [21,30]. 

A careful preoperative planning is needed and includes precise anatomical characterization of the primary tumor (unifocal/multifocal location; extent to adjacent structures) as well as a proper perioperative medical preparation [21].

Preoperative knowledge of the genetic status, in addition to other clinical variables, may affect the chosen surgical approach.

As to PCCs, resection may be open or laparoscopic, since recurrence rates do not differ among these two approaches [21]. Minimally invasive total adrenalectomy represents the preferred surgical standard over adrenal sparing surgery in cluster 1 tumors since these neoplasms harbor a high risk of recurrence and metastatic spread, especially if SDHB mutated [11,21,25,43] and in cluster 3 tumors [25].

On the contrary, in cluster 2 PCCs, cortical sparing surgery can be considered as the favored approach as it does not relate with decreased survival despite onset of recurrent disease in 13% of patients with germline RET and VHL mutations [12,25,44]. 

As to localized PGLs, surgical resection also remains the main treatment, when feasible.

The risks related to surgery vary according to the site and it is influenced by the close proximity or involvement of large vessels (internal carotid artery, aorta, mesenteric or renal arteries) and other anatomical structures (cranial nerves in case of HNPGLs) as well as by the typical rich blood supply [21,25].

Among all the PGLs, surgical treatment of HNPGLs is particularly challenging. Up-front surgery should be considered as first option in young patients, who often harbor SDHx mutations, especially in the SDHB gene, being at higher risk of metastatic spread with tumor growth [11,45]. Additional instances where surgery should be considered include rapidly progressive symptoms, or in rare cases of refractory secreting tumors or suspected malignancy [11]. However, in most cases of HNPGLs, watch and wait strategies or/and non -surgical treatment options may be preferred, especially in elderly or frail patients, in case of minimal or no symptoms and in those with multicentric lesions [11,21,46,47,48].

In advanced and metastatic setting, surgical resection of the primary tumor or metastases might be considered cautiously on a case-by-case basis [11,30,49,50], as it may improve symptoms related to CMN secretion or tumor mass effects.

However, the impact of surgical removal of the primary tumor in metastatic disease on Overall survival (OS) still remains controversial, although recent studies suggested that it may improve OS [49,50]. Of note, Gonzalez et al. demonstrated that, besides a better symptoms control after surgical resection of the primary tumors, metastatic PPGLs patients treated with surgery had longer median OS than those not treated surgically regardless of their characteristics as well as tumor features [50].

In patients with functional PPGLs, both localized and metastatic, who undergo surgery, an adequate perioperative medical management is mandatory in order to prevent potentially life-threatening events: anesthesia, tumor manipulation, previous embolization or biopsy during surgery, may determine excess of CMNs secretion, leading to hyperadrenergic symptoms and hypertensive crises [2,14,21].

### 5.2. Systemic Therapy

The management of metastatic PPGLs is challenging as there are only practiced standards of systemic therapy, that have been extensively adopted outside of controlled clinical trials [10,11,21,51,52,53,54]. To date, the only FDA-approved treatment option in metastatic disease is HSA 131 MIBG in the US.

The choice of the first line treatment and of subsequent therapy approaches depends on the patients’ characteristics, clinical symptoms, tumor type, radioisotope tumor avidity, disease burden (limited/extensive) as well as its aggressivity (limited stable disease, extensive disease with slow/rapid progression) and available resources [11,14,21,25,54].

The same state of the art recommendations currently apply to all clusters.

Although a growing number of data suggest that these different molecular subtypes may show a unique and personalized profile of sensitivity to specific drugs, a cluster specific approach has not yet entered clinical routine practice [11,12].

Most of data that correlate therapy efficacy with mutational status focus on cluster 1 tumors, particularly the most aggressive SDH-mutated ones [11]. Less is known about cluster 2 tumors as systemic treatment is infrequently necessary since only around 2–4% of metastatic PPGLs belong to this cluster.

#### 5.2.1. Chemotherapy

Chemotherapy should be considered as the first line treatment in patients who have no significant uptake of radiotracers or with clear rapid disease progression, associated with high tumor burden, or with severe uncontrolled symptoms due to CMNs secretion or mass effect, in order to obtain disease control and/or symptom palliation [10,11,30,54,55].

The standard of care, despite the lack of prospective evidence, is still represented by the combination of cyclophosphamide, vincristine, and dacarbazine (CVD), first proposed by Keiser et al. in 1985 [56]. To date, this is one of the most established and longest studied approaches in aggressive and rapidly progressive PPGLs [57,58,59,60] and it demonstrated to be particularly effective in cluster 1 SDHB-mutated patients [58,59,60].

In the largest single-institution experience with chemotherapy (not only CVD regimen) involving 54 patients, 33% of patients experienced a response defined as improved blood pressure control and/or reduced tumor size with statistically significant longer OS compared to non-responders (6.4 years vs. 3.7 years, respectively). All responders had been treated with dacarbazine and cyclophosphamide, plus at least another additional drug: vincristine was included for 14 responders while doxorubicin for 12 responders [57].

Of note, none of the 4 patients (out of 9 tested) who harbored a mutation in the SDHB gene, responded to chemotherapy. However, one patient SDHC-mutated was in the group of responders [57].

In a systematic review of four different retrospective studies involving 50 patients with metastatic PPGLs treated with CVD, Niemeijer et al. reported an overall response rate (ORR) of 41% with a disease stability achieved in 14% of cases and a biochemical response rate of 54% [58]. Interestingly, in the study of Huang which was included in this metanalysis, 9 out of 18 evaluated patients harbored a presumed mutation in SDHB/D genes and one had a confirmed SDHD mutation [59].

In 2018, Jawed et al. demonstrated that a prolonged CVD regimen (median of 20.5 cycles) was particularly effective in 12 metastatic PPGL patients harboring SDHB mutations/polymorphisms, further supporting its use in this population [60].

Temozolomide (TMZ), an orally administered alternative to intra-venous dacarbazine, showed similar efficacy in patients harboring SDHB mutations, as a first line or second line option. This evidence may rely on the hypermethylation of the O(6)-methylguanine-DNA methyltransferase (MGMT) promoter and consequent low MGMT protein expression which occur in SDHB-related tumors, thus rendering them particularly sensitive to this alkylating agent [10,11,21].

Five partial responses (33%) as well as longer PFS (19.7 vs. 2.9 months) were described in SDHB mutations carriers treated with temozolomide (150–200 mg/m^2^/day d1-5 q28) in a retrospective study of 15 patients (10 carried a germline mutation in SDHB gene) who received TMZ as first line cytotoxic chemotherapy [61].

These results outline that TMZ may represent a reasonable first line alternative to CVD in less aggressive cases or in frail patients, particularly in SDHB mutated patients.

Successful outcomes were also reported in SDHB-mutated patients treated with TMZ metronomic schedules and high-dose lanreotide as second line option, [62]. This supports its efficacy at reduced dose, particularly in SDHB mutated patient, when the standard dose is poorly tolerated.

A phase II trial (NCT04394858) investigating the synergistic effect of adding olaparib to TMZ is currently undergoing, based on the preclinical evidence of a higher activity of the PARP DNA repair system in SDHB mutated PPGLs as well as a major therapeutic effect of TMZ in mice with SDHB knockdown PPGL allograft treated in combination with Olaparib.

#### 5.2.2. Targeted Therapy

##### Tyrosine Kinase Inhibitors [TKIs]

Being altered angiogenesis a key event in the pathogenesis of several PPGLs particularly in cluster 1 and some cluster 2 (RET-driven) tumors, a growing number of studies have explored the use of TKIs in this disease during the last years [11,63].

Sunitinib was the first inhibitor proposed for the treatment of metastatic PPGLs and also the best studied in this disease [10,11,12,21,30,63].

The phase II SNIPP trial evaluated sunitinib (50 mg daily for 4 weeks followed by 2 weeks off treatment corresponding to one cycle) in 25 progressive PPGL patients showing a disease control rate (DCR) of 83%, and a mPFS of 13.4 months. Although the ORR was low in the overall unselected populations, the 3 PR occurred were observed in patients harboring germline mutations in SDHA, SDHB and RET respectively, thus suggesting that sunitinib could be a viable therapeutic option in selected patients with cluster 1 or 2 tumors [64].

The FIRSTMAPP trail (NCT01371201), the first randomized placebo-controlled trial ever conducted in PPGLs, has given sunitinib a more robust indication in progressive metastatic PPGLs, as a possible second line option. The primary endpoint was met with a PFS at 12 months of 35.9% vs. 18.9% in the sunitinib arm compared to placebo [65]. However, it still remains to be seen from the final detailed data if patients bearing SDHB mutations are really the best candidates for sunitinib [66].

Pazopanib, another TKI, showed moderate efficacy in a small PPGL cohort (n = 7, with 6 evaluable patients) with median PFS of 6.5 months and OS of 14.8 months. However, only one patient experienced a confirmed PR and common severe toxicities were observed such as hypertension (3/6) and Takotsubo cardiomyopathy (2/6) [67].

Axitinib is now being evaluated in a phase II trial in metastatic or locally advanced PPGLs (NCT03839498). Preliminary data of another phase II study (NCT01967576) investigating this drug in a similar setting showed its moderate efficacy: [11,68].

Recently, results of a retrospective study evaluating lenvatinib in 11 patients with metastatic or advanced unresectable PPGLs have been published: a mPFS of 14.7 months and an OS at 12 months of 80.8% (median not reached) were registered. Among the 8 patients with measurable disease (out of the 11 patients included), ORR was 63% (5/8 patients had a PR). Despite the potential risk of worsening hypertension, lenvatinib may be a viable treatment option for mPPGLs, Larger multicenter studies are needed to better define its potential utility in this setting [69].

As an alternative to sunitinib, targeting c-Met through cabozantinib may be of particular relevance in metastatic PPGLs, since activating mutations of the MET gene have recently been described. The TKI cabozantinib is currently under investigation for PPGL patients in a small prospective clinical phase 2 study (NCT02302833) with promising preliminary results (PR: 37%, SD: 55%, DCR: 92%, PFS: 16 months). Noteworthy, responders included *SDHB*-mutant patients [66,70,71]. This is in accordance with the results of preclinical studies on human PPGL primary cultures that showed significantly stronger efficacy of cabozantinib in cluster 1 tumors, particularly SDHB-mutated, compared with cluster 2 tumors [66,72].

Besides the extrapolation from other tumors management, the combined use of TKI plus immune checkpoint inhibitors might have a plausible rationale even in the treatment of metastatic PPGLs. Based on the high expression of PDL-1/2 reported in resected PPGLs, particularly in those belonging to cluster 1, TKIs may promote a vascular stabilization of the micro-environment that facilitates immune-response [73].

A phase II, prospective clinical trial (NCT04400474) is now assessing the role of Cabozantinib plus atezolizumab (CABATEN) in advanced and progressive tumors from endocrine system, including PPGLs and may provide important clinical data. Results are awaited.

Last but not least, two phase 2 trials studying the TKI anlotinib in advanced PPGLs are now recruiting (NCT04860700, NCT05133349).

Given the side effects of TKI treatment (e.g., nausea, vomiting, diarrhea, skin rash, hypertension, QTc prolongation), close monitoring of patients and adequate supportive therapy are essential.

##### mTORC1 Inhibitor Everolimus

Hyperactivation of kinase activity (RAS/RAF/ERK or PI3K/AKT/mTOR pathways) is a common finding in patients with cluster 2 PPGLs, thus offering the rationale for the use of mTORC inhibitors.

Growing evidence from translational studies in human PCC models, including primary tumor cultures, suggested that everolimus alone but particularly in combination is potentially effective in cluster 2-related tumors while it may be less effective in cluster 1-related disease [11,74].

Despite the robust translational evidence, only few and contradictory data derive from both a small prospective phase 2 clinical trial (DCR of 71%; genetic background not assessed) and a small retrospective study (DCR of 25%; no known SDHx mutation included) which evaluated everolimus alone in metastatic PPGLs patients [11,75,76,77].

The disappointing results of everolimus alone in vivo may be explained by the physiological activation of compensatory short-term mechanisms of resistance in response to mTORC1/p70S6K inhibition [74].

As already demonstrated in vitro, in vivo combination strategies may be useful to overcome resistance. Although sunitinib/everolimus combination treatment seemed to be less promising than other combining strategies in vitro, Ayala-Ramirez et al. reported the case of a 20-year-old woman with a *SDHB*-mutated tumor who experienced a long-term disease control under the mTORC1 inhibitor sirolimus in combination with sunitinib [11,73,78].

Altogether these data point out that it may be worth studying combined targeted therapies based on mTORC inhibitors even in cluster 1-related tumors, although they are more likely to be effective in cluster 2-related tumors. A clinical trial is needed to evaluate this therapeutic combination.

#### 5.2.3. Immunotherapy

Although PPGLs are associated with the highest rate of single germline mutations of any oncological disease, they present some of the lowest rates of somatic mutations among tumors.

Furthermore cluster-1 mutations, related to tumor “pseudohypoxia” under normoxic conditions, have always been claimed to prevent immune system recognition by interfering with T cell effector function, impairing tumor infiltration, activating immune-suppressive monocytes and increasing the expression of PDL1 and its receptor, thus resulting in immune suppression and tolerance [10,79,80]. Of note, a substantial number of apparently sporadic PPGLs may show a similar molecular profile as those that carry germline mutations of SDHx gene [81].

Altogether these data may explain why PPGLs have generally been expected to be associated with low immunogenicity and no or minimal inflammation or infiltration of T cells [81].

However, the results of a recent small prospective phase 2 clinical trial (NCT02721732) evaluating pembroluzimab in rare solid tumors including metastatic PPGLs suggested that immunotherapy may be a viable strategy in selected PPGLs patients with no other remaining therapeutic options [81,82].

Of note, in the PPGL cohort (11 patients) 4 patients achieved the primary endpoint, defined as the non-progression rate at 27 weeks after the treatment had started. Interestingly, two SDHx-mutated patients were included: one harboring a mutation in SDHD gene showed a prolonged stable disease while the other one who carried a SDHB alteration, despite a rapid substantial reduction in tumor size, did not meet the primary study endpoint since the response was not maintained due to a long recovery for severe liver toxicity [81]. Despite being interesting as well as promising, these data are too scant to assume a unique benefit for cluster 1 SDHB/D mutation carriers, as previously postulated [11,81].

It is well worth pointing out that the positive responses observed seemed to be independent of genetic backgrounds, PDL-1 expression or the presence of infiltrating mononuclear inflammatory cells in the primary tumor [81,82]. This highlights the urgent need to better define hypothetical correlations between genetic cluster affiliation and therapeutic response even in immunotherapy [11].

A phase II trial (NCT02834013), evaluating the combination of nivolumab plus ipilimumab (arm I) vs. nivolumab alone (arm II) in patients with rare tumors, including PPGLs, is currently undergoing [10]. Results are awaited.

Further investigation is needed into other factors that may contribute to the success or failure of immunotherapy, such as exposure to non-self-antigens microbially derived that mimic human cell antigens in PPGLs or combinations with other systemic therapies, such as TKIs, as already mentioned above.

In this regard, a phase 1/2, first-in-human study (NCT04187404) to assess the safety, tolerability, immunogenicity, and preliminary efficacy of EO2401, an innovative cancer peptide therapeutic vaccine in combination with nivolumab, for treatment of HLA-A2 positive patients with locally advanced or metastatic adrenocortical carcinoma or metastatic PPGLs has recently started recruiting and is undergoing.

#### 5.2.4. Cold Somatostatin Analogs (Biotherapy)

“Cold” somatostatin analogs (SSAs), octreotide LAR and lanreotide autogel, have been established as a cornerstone of the anti-proliferative therapy for midgut and pancreatic NETs. Interestingly, several PPGLs overexpress SSTRs, mainly SSTR2, particularly cluster 1 SDHx-related PGLs, which also show best responses to PRRT [10,83,84].

It could be hypothesized that SSAs may decrease HIF-α expression in PPGL cells, as already shown in NET cells. This may explain why they can be particularly effective in PPGLs with disruption of the Krebs cycle such as SDHx-related PPGLs in which a constitutive activation of the HIF pathway is described [11,85].

Although few case reports on clinical stabilization of metastatic PPGLs under SSAs have been published, there is still a lack of prospective or retrospective studies to either strengthen or refute these claims [84,86,87,88,89,90,91].

Similarly to PRRT, functional imaging with 68Ga-SSAs may help to select the patients who may benefit from cold analogs, which may represent a viable alternative in metastatic PPGLs, particularly those of the head and neck region and SDHx mutated, when PRRT is not feasible due to its limited availability and high costs.

A recent a retrospective study by Fischer et al. reported an overall DCR of 100% with first-line “cold” SSAs (6 patients, 3 SDHx-mutated), while of 67% with first-line PRRT (22 patients, 11 SDHx mutated) which increased to 73% in the SDHx subgroup. Interestingly, the authors showed that SSTR2 positivity was not only significantly associated with SDHB-and SDHx-related PPGLs, but with metastatic disease regardless of SDHB/SDHx mutation status [85].

As SSAs are well tolerated, they may represent a viable therapy option for slowly progressing metastatic PPGLs, if larger studies confirm their efficacy.

A phase II prospective study (NCT03946527), the LAMPARA trial, is currently ongoing to assess the role of lanreotide in patients with advanced disease.

However, on an individual case-by-case decision, especially when PRRT is not feasible, they may already be considered in clinical practice as an effective therapeutic strategy with the aim of disease stabilization.

### 5.3. RadioLigand Therapy [RLT]

According to the most recent guidelines, RLT can represent a valid first line therapeutic option in patients with slowly to moderately progressive disease and documented tumor uptake of the corresponding radioisotope [21,30].

The rationale for this approach relies on the evidence that around 80% of metastatic PPGLs express on their cell surface higher density of SSTRs, particularly SSTR2, or norepinephrine transporter than normal cells [85,92,93].

Overexpression of SSTR2 and norepinephrine transporter exerts a critical role in the uptake of radiolabeled SSAs (177Lu or 90Y-SSAs) and of 131I-MIBG respectively, making them a theragnostic target for the treatment of metastatic PPGLs [85,93].

Functional imaging studies with 123I-MIBG and/or radiolabeled SSAs should be performed in order to assess the affinity of the tumor cells for the radiotracer and to choose the most proper radiopharmaceutical for each case [94,95].

Patients with 123I-MIBG-avid metastatic PPGLs (less likely positive in cluster 1 SDHx-mutated tumors) may benefit from conventional 131 I-MIBG or High-Specific-Activity (HSA) 131 I-MIBG therapy, while patients with high uptake of radiolabeled SSAs (particularly cluster 1 SDHx-mutated PPGLs) may benefit from PRRT with 177Lu or 90Y-DOTA-SSAs.

On the basis of two trials, HSA I-131 meta-iodobenzylguanidine (HSA I-131 MIBG, AZEDRA^®^) was FDA- approved in July 2018 for the treatment of patients aged 12 years and older with I-MIBG–avid unresectable, advanced localized or metastatic PPGLs who require systemic therapy [21,30,96,97,98,99].

This is the first and is currently the only FDA–approved drug to treat unresectable, locally advanced, or metastatic PPGLs: prior treatment paradigms in this setting included the research and compassionate use of conventional low-specific-activity I-131 MIBG.

In the phase 2 trial (NCT00874614) the primary endpoint (reduction in baseline antihypertensive medication by at least 50%, lasting for at least 6 months) was met by 25% of all patients who received at least one therapeutic dose and 32% of patients who received 2 therapeutic doses. Among 64 patients with evaluable disease after at least one therapeutic dose, 59 (92%) had a PR (23%), or stable disease (69%), as the best objective response [97,98,99].

HSA I-131MIBG showed higher efficacy and safety compared to I-131MIBG at low-specific-activity since it consists almost entirely of 131 I-labeled MIBG and contains a significantly smaller amount of un-labeled drug thus reducing the frequency of life-threatening side effects during or shortly after drug administration related to abnormal increase of circulating catecholamines [100,101,102,103,104].

According to the most recent ESMO guidelines, although EMA has not approved HSA-131I MIBG yet, 131I-MIBG could be considered as first line approach in patients with unresectable or metastatic PPGLs with slow to moderate progression and/or high tumor burden at baseline displaying avid uptake of 123I MIBG in all tumoral lesions (evidence III, A) [30].

An alternative viable approach in the same setting, especially for cluster 1 SDHx-related PPGLs, is PRRT which use SSAs labeled with isotopes delivering a cytotoxic radionuclide [30].

PRRT was firstly approved by FDA for the treatment of GEP-NETs that are somatostatin receptor-positive [105].

Even though evidence for its use in metastatic PPGLs with high uptake of 68Ga-labeled SSA is limited (V,B according to 2021 ESMO guidelines), several small retrospective and few prospective studies suggest a high efficacy in metastatic PPGLs, especially in SDHB-mutated tumors [21,30,106,107,108,109,110,111].

Furthermore, although all radiopharmaceuticals share some common side effects, such as fatigue, toxicity to the kidneys and liver, nausea, vomiting, and low blood counts, PRRT carries a definitely lower risk of toxicity, particularly haematological, than both HAS 131I-MIBG and conventional 131I-MIBG therapy [106].

In 2011, a prospective study by Imhof et al. evaluating the efficacy and safety of 90Y-DOTA-TOC in metastasized neuroendocrine cancers showed a particularly long OS in metastatic paraganglioma patients (n = 28), thus suggesting a high therapeutic potential of PRRT in this orphan disease [107].

In 2017, Kong et al. [106] evaluated the effectiveness of PRRT in controlling hypertension in 20 consecutive retrospectively reviewed patients with advanced or metastatic PPGLs and high SSTR expression, who received 177Lu-DOTATATE PRRT with at least 3 months of follow-up from completion of treatment. Indication for PRRT was uncontrolled secondary hypertension in 14 patients and non-functional progressive metastatic disease or recurrence in 6 patients. Among the 14 patients who were given 177Lu-DOTA-octreotate treatment for uncontrolled symptoms, 8 required a lower dosage of antihypertensive medication after treatment, 5 did not need any changes in their medication dosage, and 1 patient was lost to follow-up. Of the14 patients evaluable for CT response, 5 (36%) had disease regression (29% partial and 7% minor response) while 7 (50%) had stable findings. The study found that the mPFS was 39 months, while the mOS was not reached with a median follow-up time of 28 months. Intestingly, of the 20 patients treated with 177Lu-DOTATATE, 7 had SDHB mutation while 1 was SDHD mutated.

In another retrospective study, Nastos K. et al. [108] compared PRRT with 90Y-dotatate or 177Lu-dotatate and (131) I-MIBG treatment in 22 patients with progressive or metastatic PPGLs, retrieved from their department’s database for the period from 1998 to 2013. Although patients who received PRRT experienced both statistically significant increased PFS and response to treatment when compared to those who received 131I-MIBG (*p* < 0.05), there was no significant difference in OS (*p* = 0.09). However, when comparing only patients diagnosed with PGLs, PFS, response to treatment and OS were all significantly higher in the PRRT treatment group.

In a recent meta-analysis published by our group in 2023, we assessed the efficacy of PRRT treatment based on 177Lu-DOTATATE and 90Y-DOTATOC in 213 patients affected by PPGLs. The primary endpoint was the correct quantification of DCR: a pooled DCR of 83% and 76% for 177Lu-DOTATATE and 90Y-DOTATOC treatments respectively was found, with a good toxicity profile [109].

Altogether these findings provide encouraging results for the effectiveness of PRRT in treating metastatic PPGLs, particularly for SDHx-related PGLs.

Currently, several clinical trials are underway in order to provide more robust evidence.

NCT03206060 is a recruiting open-label, single-arm, multi-center phase 2 trial evaluating efficacy and safety of Lu-177-DOTATATE in both SDHx-mutated and “apparently sporadic” progressive, SSTRs positive PPGLs.

NCT04711135 is an ongoing study assessing the safety and dosimetry of Lutathera in adolescent patients (12 to <18 years old) with SSTRs positive GEP-NETs and PPGLs.

Finally, a phase I/II study (NCT 02592707) evaluating safety, tolerability, biodistribution, dosimetry and preliminary efficacy of satoreotide tetraxetan (177Lu-IPN01072) in unresectable SSTRs positive GEPNETs, lung carcinoids, and PPGLs is currently underway. This study could provide an additional research perspective to identify therapeutic options for PPGLs.

#### 5.3.1. Future Perspectives Cluster-Oriented

Although cluster-specific therapy of metastatic PPGLs has not yet entered clinical routine practice, the distinctive molecular pathology suggests that some therapeutic options may be more effective than others in a particular cluster, as already shown in some retrospective clinical studies [11].

Potentially interesting novel targeted therapy approaches for PPGLs must be even more cluster-oriented, especially for cluster-1 and cluster-3 related tumors which are more frequently metastatic.

Especially for cluster 1-related tumors, new perspectives under evaluation include HIF-2α inhibitors, histone deacetylase (HDAC) inhibitors, DNA demethylating agents, D2 receptor antagonists and, as already mentioned above, PARP inhibitors (particularly in combination with TMZ) [10,11] (Table 1).

Belzutifan, a second-generation HIF-2α inhibitor, which has already demonstrated its efficacy among patients with VHL-associated renal cancer is currently being evaluated in an open-label, single-group phase II trial (NCT04924075, MK-6482-015) in patients with advanced or metastatic PPGLs with adequately controlled blood pressure (ORR as primary endpoint) [112].

Another HIF2α inhibitor, DFF332, as single agent and in combination with everolimus or spartalizumab, plus the adenosine A2A receptor antagonist taminadenant is currently under investigation in a phase 1 trial in tumor patients with HIF-stabilizing mutations, including PPGLs (NCT04895748).

Recently, the promising results of a phase II trial (NCT03034200) evaluating an oral selective antagonist of the dopamine D2 receptor (ONC201) in neuroendocrine tumors and desmoplastic small round cell tumors (DSRCT) have been published, showing the greatest benefit especially in metastatic PPGLs. Of 10 patients with metastatic PPGLs enrolled, 50% (5/10; 3 SDHB-mutated) exhibited a PR and 2 patients (1 SDHB-mutated) had SD for more than 3 months, with good tolerability [113].

Despite the typical high expression of D2 receptors in PPGLs [114], discerning whether DRD2 antagonism, ClpP agonism or both contributed to responses in metastatic PPGLs is not easy in the absence of correlative studies on tissues [113,115].

Furthermore, given the SDHB mutational status of 4 responders and the ability of another imipridone compound with similar profile to downregulate SDH-A/B in other cancer preclinical models, it could be interesting to evaluate a possible enhanced efficacy of ONC201in SDH deficient tumors. This may shed light on new potential biomarkers, such as DRD2 expression, ClpP status, *SDH* x mutations, or some combination of them, to explore in metastatic PPGLs [115].

As to cluster 3-related PPGLs, targeting Wnt signaling seems to be a reasonable option. Of note, both the PORCN inhibitor WNT974, which inhibits Wnt signaling, and the ß-catenin inhibitor PRI-724 showed good efficacy in several NET cell lines, thus shedding light on a new potential therapeutic target in metastatic PPGLs [11,116].

Although cluster 2-related PPGLs show a very low metastatic risk, it is well worth mentioning that a phase 2 pediatric trial (MATCH, NCT04284774) evaluating tipifarnib, a farnesyl-transferase inhibitor that disrupts HRAS function, in patients with *HRAS*-mutant tumors, including pheochromocytomas, is currently recruiting and may provide important data for PPGL management (Table 2).

#### 5.3.2. Locoregional Procedures

The greatest experience with conventional external beam radiotherapy (cERBT) or stereotaxic radiosurgery derive from the treatment of HNPGLs, where these approaches have been used as alternatives to surgery for locations with high surgical risk (carotid or intracranial involvement) or when patients are not candidates for surgery (elderly and/or frail patient, minimal or no symptoms, bilateral and/or multifocal tumors) [11,21,48,117].

In oligo-metastatic PPGLs, locoregional approaches including radiotherapy, stereotaxic surgery, radiofrequency ablation or cryoablation, percutaneous ethanol injection, may represent reasonable options for local control and managing mass effect symptoms or hormone-related symptoms [11,21,37,118,119,120]. However, indications must be individualized and discussed within multidisciplinary team as no prospective data are available.

## 6. Follow-Up Recommendations

Ten-year follow-up is recommended for all patients with resected PPGLs, as they are all considered at risk of tumor recurrence. However, it must be intensified and extended to the whole life in patients with hereditary forms or considered to be at high risk of metastatic disease [21]. Currently, no specific follow-up protocols are established: monitoring strategies must include regular biochemical testing as well as imaging studies. Whenever possible, such follow-up should be performed by multidisciplinary team at a tertiary center [21].

## 7. Conclusions

PPGLs are rare, complex and still little-known clinical entities in the scientific research landscape.

Early detection, genetic testing, and a multidisciplinary approach are of utmost importance for a proper management. Surgery, if feasible, must be the first-line treatment for localized PPGLs as it represents the only curative option.

In advanced and/or metastatic settings, several therapeutic options are available: the choice among them must be tailored to the unique characteristics of the patient and the tumour considering the benefits and risks of each option. However, only few of them hinge upon prospective data.

Several clinical trials prospectively assessing the role of strategies which have already been adopted in research and compassionate programmes, evaluating new combinations, and testing novel targeted therapy approaches, even more cluster-oriented, are currently underway. Results are awaited.

Based on the growing preclinical evidence, it is undoubtable that genetic testing for germline mutations as well as genome profiling for somatic mutations, where available, must be improved and become standard practice in order to render management even more cluster-specific as well as personalized and improve the prognosis of this rare and molecularly complex disease.

## Figures and Tables

**Figure 1 pharmaceuticals-17-00354-f001:**
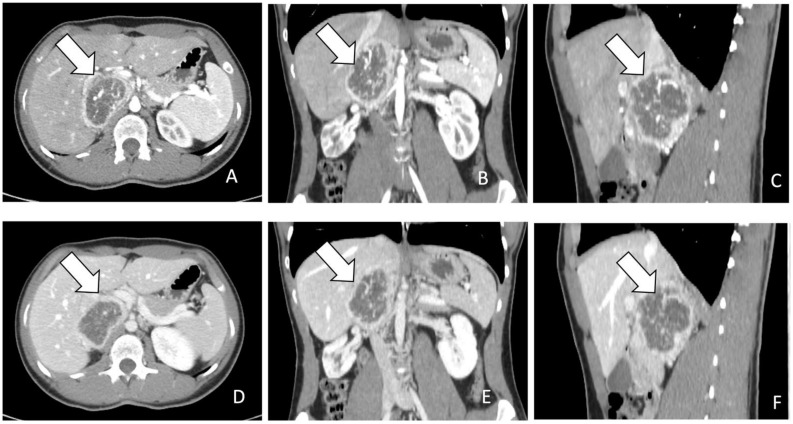
(**A**–**F**): (**A**). CT images in axial plane during arterial phase of contrast study. The lesion indicated by the arrow appears as a heterogenous mass with rim enhancement. (**B**,**C**). MPR post processing in coronal and sagittal planes during arterial phase. (**D**). CT images in axial plane during portal phase of contrast study. The lesion indicated by the arrow appears heterogenous with rim enhancement and without progressive contrast enhancement. (**E**,**F**). MPR post processing in coronal and sagittal planes during portal phase.

**Figure 2 pharmaceuticals-17-00354-f002:**
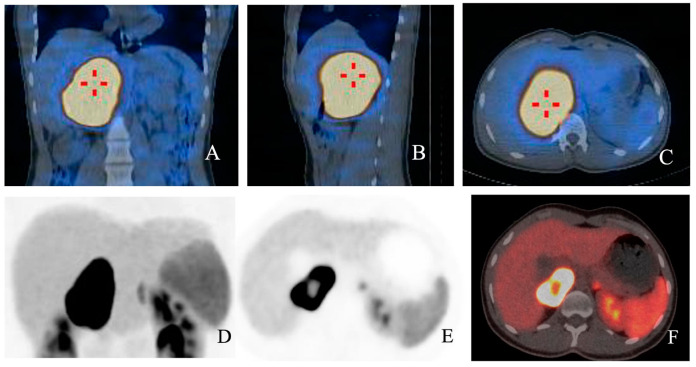
(**A**–**F**): Fusion images with 123I MIBG SPECT/CT in coronal (**A**), sagittal (**B**) and axial planes (**C**) respectively showing high radioligand uptake in a young patient diagnosed with PCC. In (**D**,**E**), functional images from the same patient show high uptake of 68Ga-DOTA-SSA, with a central cold area. In fusion image (**F**), a central spared area of necrosis is present.

**Table 1 pharmaceuticals-17-00354-t001:** Main characteristics of PPGLs according to molecular clusters [10,11,12,13,14].

	Genes	Mutations	Biochemical Phenotype	Most Sensitive Functional Imaging Modality	Metastatic Risk
Cluster 1	Krebs cycle-related genes: SDHx, SDHAF2 FH, MDH2, IDH, SLC25A11	predominantly germline	Predominantly noradrenergic and/or dopaminergic	68 Ga DOTA SSA PET/CT	High-intermediate
	Pseudohypoxia VHL/EPAS 1 related-genes: VHL, EPAS1/2, PHD1/2, IRP1	either germline and somatic	noradrenergic	[18F] F DOPA PET/CT	Intermediate
Cluster 2	Kinase signaling: RET, NF1, MAX, HRAS, TMEM127	both germline and somatic	typical adrenergic (epinephrine/metanephrine-E/MN-> both E/MN and NE/NMN)	[18F] FDOPA PET/CT	Low
Cluster 3	Wnt signaling: MAML3, CSDE1	somatic	unknown	unknown	High-intermediate

**Table 2 pharmaceuticals-17-00354-t002:** Summarizes the possible novel therapeutic approaches under evaluation in ongoing trials.

Trial	Phase	Therapy	Disease	Status
NCT04924075	II	Belzutifan	Advanced PPGLs+ pancreatic Neuroendocrine Tumor (pNET), both unselected for germline VHL mutations and selected (+other tumors)	Recruiting
NCT04895748	I/Ib	DFF332 alone and in combination with everolimus or spartalizumab, plus taminadenant	Advanced and/or recurrent Clear cell renal cell carcinomas and other malignancies with HIF stabilizing mutations, including PPGLs	Recruiting
NCT04394858	II	Olaparib+ temozolomide vs. temozolomide alone	Advanced and/or metastatic PPGLs	Recruiting
NCT04284774 (pediatric)	II	Tipifarnib	Metastatic HRAS-mutant tumors, including PCCs	Recruiting

## Data Availability

Not applicable.

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
