# Peer review of "The Management of Phaeochromocytomas and Paragangliomas in the Era of Precision Medicine: Where Are We Now? Evidence-Based Systemic Treatment Options and Future Cluster Oriented Perspectives"

_pharmaceuticals, 2024, doi:10.3390/ph17030354_

Round 1
Reviewer 1 Report
Comments and Suggestions for Authors
I would like to thank the editor for letting me review this interesting manuscript on PPGL.
The manuscript in its current form contains 8 sections of heterogeneous quality. My advice would be to revise the framework of the manuscript to enhance its understanding.
'Introduction' (part 1) is well written.
'Clinical presentation and metastatic potential' (part 3) and 'Diagnosis and staging' (part 4) could be part of the same section, using subsections.
In this section :
Figure 1 shows CT images and MRI images could be added to inform morphologic examinations. (A) and (B) images need cropping.
Figure 2 and 3 could be gathered to inform functional imaging. Figure 2 images need cropping. Images form the same patient using different radioligand would be useful.
'Genetics ands molecular classification...' (part 2) is a interesting section. My advice would be to use subsection and to inform on the differences between clusters regarding biochemical phenotypes, functional imaging, prognosis...
The role of dynamic contrast-enhanced MRI (DCE-MRI) should be added.
In addition, adding a table comparing clusters would be helpful to readers.
Limitations regarding genetics and molecular classifications (long waiting period for diagnosis) should be added.
Therapeutic options (part 5) also needs framework revising. After description of therapeutic options, authors should try to organize this section on the role of each therapeutic option according to disease stage and cluster-oriented therapeutic approaches whenever appropriate. This would suit best the proposed title of the manuscript.
'Surgery' subsection needs revising. The role of interventional radiology should be added.
'Prognosis' (part 6) and 'Follow-up' (part 7) could be allocated to previous sections.
'Conclusion' (part 8) is appropriate.
Comments on the Quality of English LanguageEnglish is understandable.
'Surgery' paragraph needs editing : "even when", "unpleasant", "plays just" are unappropriate.
TMZ ... "showed to be efficacy" needs correction.
Author Response
thanks for the comments We have tried to make all the suggested changes Best regardsReviewer 2 Report
Comments and Suggestions for Authors
Interesting and well presented paper-review; I would suggest to implement and up-to-date the sub-paragraph entitled "Diagnosis and Staging" with diagnostic imaging informations regarding atypical as well as malignant pheochromocytomas (Pheos); in this regard, the diagnosis of atypical Pheos may be not easy and frequently misleading as well as the diagnosis of malignant Pheos should be early to have a favourable prognosis or for advanced cases the post-treatment follow-up require the appropriate imaging evaluation; these issues should be discussed and for this purpose I suggest the following references:
1. Galatola R, Romeo V, Simeoli C, Guadagno E, De Rosa I, Basso L, Mainolfi C, Klain M, Nicolai E, Colao A, Maurea S, Salvatore M. Characterization with hybrid imaging of cystic pheochromocytomas: correlation with pathology. Quant Imaging Med Surg. 2021 Feb;11(2):862-869. doi: 10.21037/qims-20-490. PMID: 33532285; PMCID: PMC7779916.
2. Maurea S, Fiumara G, Pellegrino T, Zampella E, Assante R, Mainenti P, Cuocolo A. MIBG molecular imaging for evaluating response to chemotherapy in patients with malignant pheochromocytoma: preliminary results. Cancer Imaging. 2013 Apr 15;13(2):155-61. doi: 10.1102/1470-7330.2013.0017. PMID: 23598367; PMCID: PMC3629891.
3. Maurea S, Cuocolo A, Imbriaco M, Pellegrino T, Fusari M, Cuocolo R, Liuzzi R, Salvatore M. Imaging characterization of benign and malignant pheochromocytoma or paraganglioma: comparison between MIBG uptake and MR signal intensity ratio. Ann Nucl Med. 2012 Oct;26(8):670-5. doi: 10.1007/s12149-012-0624-1. Epub 2012 Jul 1. PMID: 22752959.
4. Galatola R, Attanasio L, Romeo V, Mainolfi C, Klain M, Simeoli C, Modica R, Guadagno E, Aprea G, Basso L, Nicolai E, Salvatore M, Maurea S. Characterization of atypical pheochromocytomas with correlative MRI and planar/hybrid radionuclide imaging: a preliminary study. Applied Sciences 2021; 11: 9666
Author Response
thanks for the comments We have tried to make all the suggested changes Best regardsReviewer 3 Report
Comments and Suggestions for Authors
The article titled "The Management of Phaeochromocytomas and Paragangliomas in the Era of Precision Medicine: Where are We Now? Evidence-based Systemic Treatment Options and Future Cluster-oriented Perspectives" provides a comprehensive review of the current evidence regarding the diagnosis and treatment of phaeochromocytomas (PCCs) and paragangliomas (PGLs), collectively known as PPGLs. The article discusses the genetic and molecular classification of PPGLs, their impact on phenotype, and the need for a more cluster-specific approach in clinical practice.
Following are a few areas for improvement of the article.
1. Limited Prospective Data: The article mentions that only a few therapeutic options for advanced/metastatic PPGLs are based on prospective data, which implies a need for more robust clinical trials to validate the efficacy of these treatments.
2. Slow Accrual in Clinical Trials: The article references several clinical trials that were halted or had to be converted into retrospective studies due to slow accrual rates. This indicates a challenge in recruiting sufficient participants for PPGL studies, which may limit the generalizability of the findings.
3. Small Sample Sizes: Some of the studies mentioned in the article had small cohorts (e.g., a study with only 7 PPGL patients), which may need to provide more statistical power to draw definitive conclusions about the efficacy of treatments.
4. Severe Toxicities: The article reports common severe toxicities observed in some of the studies, such as hypertension and Takotsubo cardiomyopathy, which raises concerns about the safety profile of specific treatments.
5. Need for Larger Multicenter Studies: The article suggests that more extensive multicenter studies are needed to better define the potential utility of specific treatments, such as lenvatinib, in managing metastatic or advanced unresectable PPGLs.
6. Lack of Cluster-Oriented Approach: Despite identifying molecular clusters that could guide treatment, a cluster-oriented approach has yet to be established in clinical practice. This indicates a gap between research findings and their application in patient care.
7. Potential Bias in Author Affiliations: All authors are affiliated with the same institution, which could introduce bias or limit the diversity of perspectives in the review.
8. Lack of Discussion on Non-Pharmacological Treatments: The focus of the article is primarily on pharmacological interventions, and there needs to be more discussion on the role of non-pharmacological treatments, such as surgery or radiation therapy, in the management of PPGLs.
9. No Clear-cut Features for Metastatic Behavior: The article states that there are no definitive features to predict metastatic behavior in PPGLs, a significant shortcoming in managing these tumors.
10. Lack of EMA Approval for Certain Treatments: The article mentions that the European Medicines Agency (EMA) has yet to approve specific treatments discussed, which may limit their availability and use in European clinical practice.
11. Inconsistencies in Predictive Scales: The article discusses various predictive scales for malignant potential, such as the PASS score and GAPP scale, but notes that the World Health Organization (WHO) neither endorses nor discourages their use, indicating a lack of consensus on their reliability.
12. Challenges in Biochemical Diagnosis: While the article describes the use of plasma-free and urinary fractionated metanephrines (MNs) for screening and follow-up of PPGLs, it does not address potential challenges or limitations in biochemical diagnosis, such as false positives or the need for confirmatory testing.
13. Limited Discussion on Radiological and Nuclear Medicine Imaging: The article briefly mentions the role of various imaging techniques in the diagnosis and staging of PPGLs but does not provide a detailed discussion on their limitations, such as false negatives or the need for multiple imaging modalities to assess the disease accurately.
14. Insufficient Evidence for Treatment Decision Making: The article acknowledges that evidence to support treatment decision-making is inferior, highlighting the need for more research to guide clinical practice.
15. Limited Data on Chemotherapy Efficacy: The article cites limited and variable evidence on the efficacy of chemotherapy regimens like CVD (cyclophosphamide, vincristine, and dacarbazine), which is still considered the standard of care despite the lack of prospective evidence.
16. Challenges with Targeted Therapies: The article discusses the moderate efficacy of targeted therapies like pazopanib and axitinib. It also notes the challenges in accruing patients for clinical trials and the severe toxicities observed, which may hinder their widespread adoption.
17. Contradictory Data on mTORC Inhibitors: The article presents contradictory data on the efficacy of mTORC inhibitors like everolimus, with some studies showing stable disease and others leading to progressive disease, indicating uncertainty about their clinical benefit.
18. Potential Ineffectiveness of Immunotherapy: The article suggests that PPGLs may be associated with low immunogenicity. This could explain the minimal inflammation or T-cell infiltration observed, potentially making immunotherapy less effective for these tumors.
19. Lack of Comprehensive Treatment Goals Discussion: While the article outlines the main goals of treatment, it needs to provide an in-depth discussion on how these goals are prioritized or achieved in practice, particularly in balancing symptom management with the risks of treatment.
In summary, while the article provides valuable insights into the management of PPGLs, it also highlights several shortcomings, including the need for more prospective data, challenges in clinical trial recruitment, small sample sizes, severe toxicities associated with treatments, the necessity for more extensive multicenter studies, the absence of a cluster-oriented approach in practice, incomplete citation information, potential bias due to author affiliations, limited discussion on non-pharmacological treatments, and the lack of predictive features for metastatic behavior.
Comments on the Quality of English Languageminor
Author Response
thanks for the comments We have tried to make all the suggested changes Best regardsRound 2
Reviewer 1 Report
Comments and Suggestions for Authors
I have no further corrections.